# Natural Language Inference over Tables: Enabling Explainable Data Exploration on Data Lakes

Mario Ramirez[1], Alex Bogatu[1], Norman W. Paton[1], and André Freitas[1,2]

[1] Department of Computer Science, University of Manchester, UK
[2] Idiap Research Institute, Switzerland
`mario.ramirezorihuela@postgrad.manchester.ac.uk`
`{alex.bogatu,norman.paton,andre.freitas}@manchester.ac.uk`

**Abstract.** Data lakes are repositories of data with potential for analysis. Data lakes aim to liberate data from silos, thereby enabling cross-cutting analyses that were hitherto out of reach. This gives rise to significant challenges for data scientists simply discovering what data sets may be relevant to a task-in-hand. Given a data set of interest, several proposals have been made for indexing schemes that can identify related data sets. However, such schemes tend to build on similarity metrics that stop short of providing a clear explanation as to how an identified data set relates to a provided target. We address this problem by applying Natural Language Inference (NLI) to providing explanations as to how the attributes of discovered data sets relate to those of the target, in terms of a collection of semantic relations. We provide two approaches to inferring semantic relations: (a) by performing unsupervised intensional and extensional analysis of the data sources using Natural Language Processing techniques; and (b) by performing supervised learning of semantic relations by applying BERT over source schema information. The contributions of this paper are: an NLI strategy for providing explicit characterisation of semantic relations between data sets; two approaches to inferring the semantic relations; and an empirical evaluation of the approaches using open government data.

## 1 Introduction

The growing availability of potentially valuable datasets is leading organisations to develop centralised, scalable repositories, such as data lakes [21]. On-demand infrastructures allow access to such data for analytics and reporting [9]. This creates a data discovery problem, identifying data sources that are relevant to an information requirement.

Commonly, the data discovery problem is formulated as the computation of a similarity function, which aggregates semantic, morphological and distributional features into a similarity score (e.g, [2], [4]). More recently, the evolution of neural language models [8] has lowered the barriers for complex language interpretation and inference. Contemporary architectures (e.g. transformer-based models) [22] have consistently delivered accurate language inference capabilities across different tasks. Additionally, variations of these models have been adapted to operate over structured (e.g, [10], [5]) and semi-structured data [24]. However, these models commonly trade interpretability

| Semantic Relation | Symbol | Abbreviation | Example |
|---|---|---|---|
| Equivalence | $\equiv$ | *EQ* | $couch \equiv sofa$ |
| Forward Entailment | $\sqsubset$ | *FWD* | $crow \sqsubset bird$ |
| Reverse Entailment | $\sqsupset$ | *REV* | $European \sqsupset French$ |
| Negation | $\wedge$ | *NEG* | $human \wedge nonhuman$ |
| Alternation | $\mid$ | *ALT* | $cat \mid dog$ |
| Cover | $\smile$ | *COV* | $animal \smile nonhuman$ |
| Independence | $\#$ | *IND* | $hungry \# hippo$ |

Table 1: Semantic Relations in NLI.

and semantic control for inferential performance, operating by the same principle of a latent space (a black-box) delivering a similarity score.

*Approach:* In this paper we aim to make explicit the nature of the relationships between tables using Natural Language Inference (NLI). NLI provides a set of atomic natural language inference computations that deliver a step-wise and transparent semantic inference model [13]. For example, NLI can recognise that the text *Kennedy was chosen* can be inferred from *JFK was elected*. Textual Entailment has been seen as a framework for modelling semantic inference that can be generalised into entailment engines for use in many applications [1]. We adapt the formal model of NLI into a variant for structured data, named Relational Natural Language Inference model (RNLI), in which we extend the NLI paradigm to exploit intensional and extensional features of data sets. Given a collection of sources, RNLI outputs a set of entailment relations between pairs of table attributes to qualitatively determine candidate sources for a given query. The specific semantic entailment relations inferred are listed in Table 1.

*Motivating Scenario.* Consider a data scientist that needs to build a report of the best universities based on their rankings, fees and potential graduate employability. Additionally, it is of interest to know factual information about safety in the cities where universities are based on. The report would be as table $H$ in Figure 1. To address this task, the data scientist has access to a large collection of data sets related to the domains of the original requirement. Therefore, one key challenge is to find candidate tables in the data lake that contain properties for the expected output. Based on the sources shown in Figure 1, the aim of the data discovery task is to determine which attributes of candidate sources $S_1$, $S_2$ and $S_3$ are relevant to the target table $H$. Specifically, it is of interest of the final user to obtain an explainable output to understand in the form of the semantic relations between source and target attributes.

The contributions of this paper can be summarised as:

1. A proposal of an interpretable semantic entailment framework for tabular datasets (RNLI).
2. An implementation of this model using a rule-based linguistic feature model and a transformer-based architecture.
3. An evaluation of the model for the task of explainable table entailment.

$S_1$: University Rankings

| Name | Location | University Rank | Tuition and Fees |
|---|---|---|---|
| Princeton University | Princeton, NJ | 1 | $45,320 |
| Cornell University | Ithaca, NY | 15 | $50,953 |

$S_2$: Graduate Employability

| University | State | City | Rank |
|---|---|---|---|
| Harvard University | Massachusetts | Cambridge | 1 |
| Columbia University | New York | Now York | 14 |

$S_3$: City Safety Statistics

| City | Rank | Safety Index |
|---|---|---|
| Memphis, TN | 1 | 24,42 |
| New York, NY | 34 | 56,3 |

$H$: University Selection Indicators

| University | Ranking | Fees | Employability Rank | City | Safety Index |
|---|---|---|---|---|---|
| Harvard University | 2 | $47,074 | 1 | Princeton | 30 |
| Columbia University | 5 | $55,056 | 14 | Now York | 60.4 |

Fig. 1: Running example: source ($S_i$) and target ($H$) tables.

This paper is organised as follows. Related work regarding the data discovery problem is described in Section 2. Section 3 formalises and describes the proposed RNLI model and the transformer–based alternative as a mechanism to deliver RNLI. Section 4 provides a qualitative and quantitative empirical analysis using large and diverse real-world data collections from open government data. Finally, conclusions are presented in Section 5.

## 2   Related Work

Data discovery approaches can be characterised on the basis of the evidence that informs the discovery process and the nature of its results. For example, Pham *et al.* [19], Das Sarma *et al.* [7] and Ventis *et al.* [23] rely on external databases, such as WebIsA [20] and Freebase [3], to perform entity annotation in tables. These approaches combine string and entity similarities to relate source and target tables. Alternatively, ontology based approaches address the discovery problem by annotating cells and headers in each of the sources [12]. However, these approaches are limited in scope by their dependence on external knowledge bases.

More recent approaches have fewer external dependencies. For example, Aurum [4] adopts a data-driven approach by using data summarization and hashing to capture relationships between sources, to create what are referred to as enterprise knowledge graphs. An extension, SemProp [5], expands the similarity framework to include semantic similarities based on word embeddings. For the most part, these approaches provide quantitative measures of similarity that are used to rank sources.

$D^3L$ [2] provides evidence–based models, where features are extracted from attribute names and values to capture similarity signals. Consequently, a common space of features is generated, from which distance vectors are used for similarity measurements. Like Aurum, $D^3L$ stops short of providing a clear explanation as to how an identified data set relates to other datasets or to a provided target.

There is, therefore, a lack of explanatory information for the identified relationships between tables/attributes in data discovery, a gap that we seek to bridge using *language inference*. In particular, we pioneer the use of NLI [14] to infer explanatory semantic relations between tables and attributes using NLP-based analysis and transformer–based [22] classification. The former builds on a Relational Natural Language Inference Model that exploits intensional and extensional features to compute similarity relationships in an explainable way.

Our second approach builds on a deep-learning transformer architecture [22], specifically BERT [8]. Bidirectional Encoder Representations from Transformers (BERT) [8] is a universal language model pre-trained on large amounts of textual data with the aim of providing a solid base model that can be further fine-tuned to accommodate different downstream tasks (e.g., we use it for NLI-based semantic tagging) in a supervised manner with relatively little additional training. BERT proposes a technique for representing sentences in a similarity-preserving Euclidean space, where semantically similar constructions are close by. We built on this similarity-preserving property to generate representations for schema-level table information (i.e., table and attribute names) and use them to classify the different types of similarity relationships they find themselves in, in accordance to the RNLI model.

## 3    Relational Natural Language Inference (RNLI)

In this paper, we show that explaining the similarity relationships existing between different tables/attributes of a data lake can be achieved using unsupervised analysis grounded on intentional (i.e., schema–level) and extensional (i.e., instance–level) evidence. Additionally, we also describe a supervised alternative, construed as a classification problem that assumes the existence of training data. The unsupervised proposal, described in this section, builds on Natural Language Inference and aims to support dataset discovery through the generation of semantic alignments between tables in data lakes by relying on various evidence types, as we now describe.

### 3.1    Semantic evidence types

Let $\mathcal{D}$ be a data lake that consists of a set of source tables $\mathcal{D} = \{S_1, ..., S_n\}$. Each $S_i \in \mathcal{D}$ is composed of a set of attributes $\{a_1, ..., a_m\}$. In identifying attribute pairs that are semantically related, we consider four types of semantic evidence:

$\mathbb{N}$ : the table descriptors (e.g., table names) of elements in $\mathcal{D}$, with $\mathbb{N} = \{\mathtt{n}_1^S, \ldots \mathtt{n}_n^S\}$ the set of table names.

$\mathbb{A}$ : the attribute descriptors (e.g., attribute names) of attributes in $\mathcal{D}$, with $\mathbb{A}_i = \{\mathtt{n}_i^1 \ldots \mathtt{n}_i^m\}$ the set of attribute names of source $S_i$.

$\mathbb{T}$ : the attribute data types (e.g., numerical, categorical, etc.) of attributes in $\mathcal{D}$, with $\mathbb{T}_i = \{\mathtt{t}_i^1 \ldots \mathtt{t}_i^m\}$ the set of attribute types of source $S_i$.

$\mathbb{D}$ : the value domains of attributes in $\mathcal{D}$, with $\mathbb{D}_i = \{\mathtt{d}_i^1 \ldots \mathtt{d}_i^m\}$ the set of value domains of attributes of source $S_i$. Here, a domain $\mathtt{d}_i^j$, is defined as a collection of representative terms, shared by the extents of attributes pertaining to $\mathtt{d}_i^j$, and identified through value extent analysis using specific domain discovery techniques, such as *D4* [18].

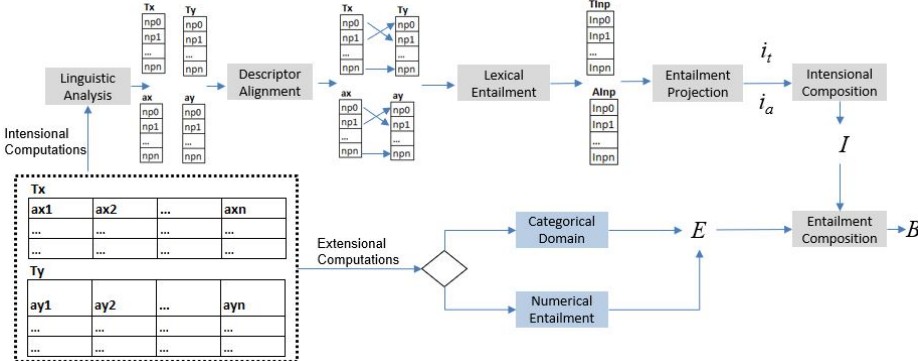

Fig. 2: Relational Natural Language Inference Model

As part of our unsupervised NLI–based approach, $\mathbb{N}$ and $\mathbb{A}$ denote *intensional* evidence types, while $\mathbb{T}$ and $\mathbb{D}$ denote *extensional* evidence types. We take a multi–evidence approach and combine both intensional and extensional types of signal to derive similarity features that have the potential of explaining the relationships beyond abstract relatedness.

### 3.2   Intensional Entailment Computation

This subsection describes how the RNLI model identifies attributes that are semantically related on the basis of intensional information, to determine the conceptual meaning of table and attribute names. We rely on the presence of natural language descriptors as part of table and attribute names and aim to capture these types of signal in the following similarity features:

***Full literal match***: Given a pair of table/attribute names, $(n_i^S, n_j^S)/(n_k^i, n_l^j)$, construed as strings $(s_i, s_j)$, we define $l^f(s_i, s_j)$, the full literal match, as a binary feature (i.e, $EQ$ or $IND$) between $s_i$ and $s_j$.

***Head term literal match***: Given a pair of table/attribute names, $(n_i^S, n_j^S)/(n_k^i, n_l^j)$, construed as strings $(s_i, s_j)$, we define $l^h(s_i, s_j)$, the head term literal match, as a binary feature (i.e, $EQ$ or $IND$) between $s_i$'s and $s_j$'s head terms. We obtain head terms from noun phrases identified using NLP–specific techniques (e.g., Part of Speech (PoS) tagging).

***Head term synonymic match***: Given a pair of table/attribute names, $(n_i^S, n_j^S)/(n_k^i, n_l^j)$, construed as strings $(s_i, s_j)$, we define $s^h(s_i, s_j)$, the head term synonymic match, as a binary feature (i.e, $EQ$ or $IND$) between the Wordnet–specific synsets of $s_i$'s and $s_j$'s head terms.

***Head term taxonomic match***: Given a pair of table/attribute names, $(n_i^S, n_j^S)/(n_k^i, n_l^j)$, construed as strings $(s_i, s_j)$ in an already determined synonymic equivalence as per $s^h$, we define $t^h(s_i, s_j)$, the head term taxonomic match, as a multi–valued feature. The value of $t^h$ is determined based on the previously identified head terms and modifiers.

Specifically, equality between the two sets of modifiers leads to $EQ$, containment leads to $FWD$ or $REV$, partial overlap leads to $COV$, and non–overlap leads to $ALT$ because the head terms already match as per $s^h$.

Given a target table $H \in \mathcal{D}$ (i.e., as in the example from Figure 1), in order to identify similar attributes to the attributes in $H$, we perform a pair–wise processing of attributes in $H$ and all the other attributes of tables in $\mathcal{D}$, and extract the above–defined $l^f, l^h, s^h$, and $t^h$, for each attribute pair[3]. The process for intensional entailment computation responsible for extracting the above features is depicted in the upper part of Figure 2 and described next.

**Linguistic Analysis.** The first step in the process is linguistic analysis of table/attribute names. We parse each table/attribute name to obtain their PoS labels in order to heuristically generate a structured representation. Using PoS tags and Named Entity Recognition (NER) we split each table/attribute name in a sequence of *noun phrases*. We extract the *head terms* of the noun phrases and use them to compute $l^h$, while $l^f$ is computed using the full table/attribute names. Often, noun phrases contain additional terms called *modifiers* that provide additional information about the concept of the phrase. We use these modifiers next.

**Descriptor Alignment.** From the collection of noun phrases associated with each table/attribute name we perform phrase alignments to simplify entailment computation. We use the head term of each noun phrase to perform a first conceptual comparison. We use Wordnet to identify synonymic relations between head terms to determine candidate alignments between table/attribute names, i.e., $s^h$. Additionally, we use the modifiers associated with each head term, when they exist, to discover a taxonomic relation between noun phrases, i.e., $t^h$. For example when, comparing *Regional University Rank - September* against *University Rank - October* we aim to align the noun phrases (*Regional University Rank, University Rank*) and the modifiers (*September, October*) to determine an entailment relation between noun phrases in terms of their taxonomic representation, as shown in Algorithm 1. In the algorithm, $isSynonymic$ determines the value of $s^h$ based on Wordnet. When $s^h = EQ$, we proceed with a lexical entailment analysis (i.e., GETTAXONOMICREL) described next.

**Lexical Entailment.** With alignments identified in the previous step we now proceed to determine lexical entailments between noun phrases using GETTAXONOMICREL. First, tokenized table/attribute names are construed as sequences of modifiers plus one head term per name. We perform comparisons between modifier tokens to check for overlapping concepts. We also consider the number of modifier tokens and assume that the more modifiers exist the more specific a concept is being represented. Then, we conclude an entailment relation, i.e., the value of $t^h$, based on the rules already mentioned in the head term taxonomic match feature definition. For instance, in the previous example, we can see that *Regional University Rank* forward entails *University Rank* and *September* is an alternation of *October*. The composition of these two different relations is described next.

---

[3] In practice, one could drastically reduce the potentially prohibitive space of attribute pairs to process by initially performing general similarity discovery (e.g., using $D^3L$ [2]) and apply RNLI only on the resulted similar pairs.

**Algorithm 1** Compute noun phrase entailment

1: *Input:* noun phrases $np_x$, $np_y$
2: *Output:* $i_{np}$ entailment relation between noun phrases
3: **function** GETNPRELATION($np_x$, $np_y$)
4:    $h_x \leftarrow x.getHeadToken()$
5:    $h_y \leftarrow y.getHeadToken()$
6:    **if** !isSynonymic($h_x$, $h_y$) **then**
7:        $i_{np} \leftarrow independence$
8:    **else**
9:        
          $i_{np} \leftarrow$ GETTAXONOMICREL($np_x$, $np_y$)
10:   **end if**
11:   **return** $i_{np}$
12: **end function**

**Algorithm 2** Compute intensional entailment composition

1: *Input:* table, attribute entailments $i_t$, $i_a$
2: *Output:* Composite Intensional Entailment $I$
3: **function** COMPUTEINTENSIONALVALUE($i_t$, $i_a$)
4: **if** $i_t = IND$ **then**
5:    **if** $i_a$ != *IND* **then**
6:        $I \leftarrow ALT$
7:    **else**
8:        $I \leftarrow IND$
9:    **end if**
10: **else if** $i_t = FWD$ or $i_t = REV$ **then**
11:    **if** $i_a = EQ$ **then**
12:        $I \leftarrow i_t$
13:    **else**
14:        $I \leftarrow i_a$
15:    **end if**
16: **else**
17:    $I \leftarrow i_a$
18: **end if**
19: **return** $I$
20: **end function**

**Entailment Projection.** Given a pair of table/attribute names, the previous steps of our approach can output multiple candidate alignments between associated noun phrases. Consequently, this leads to multiple possible values for $t^h$. We aggregate such cases according to the rules in Figure 3. For instance, in the previous example, Algorithm 2 determines that *Regional University Rank - September* is an alternation of *University Rank - October*.

**Entailment Composition.** The intensional features described at the beginning of this section are extracted separately for each pair of table names and for each pair of attribute names. In considering both table– and attribute–level intensional evidence, we assume that attribute names provide specific information of a concept which requires a contextualisation from the table name. Algorithm 2 is used to compute an intensional entailment by composing table and attribute level entailments. This is a rule-based algorithm in which, based on the table–level and attribute–level intensional entailments, a final intensional entailment is produced.

### 3.3  Extensional Entailment Computation

We now describe how we extract extensional features from the value extent of each attribute in a given table. We extract three types of extensional similarity features:

*Data type match*: Given a pair of attribute extents, ($[\![a_i]\!]$, $[\![a_j]\!]$), we define $q^v([\![a_i]\!], [\![a_j]\!])$, the data type match, as a binary feature (i.e, $EQ$ or $IND$) between ($[\![a_i]\!]$ and $[\![a_j]\!]$). We use a simple classification of data types: *categorical* or *numerical*.

| ○ | ≡ | ⊏ | ⊐ | ⋀ | ⎮ | ⌣ | # |
|---|---|---|---|---|---|---|---|
| ≡ | ≡ | ⊏ | ⊐ | ⋀ | ⎮ | ⌣ | # |
| ⊏ | ⊏ | ⊏ | ⌣ | ⌣ | ⎮ | ⌣ | # |
| ⊐ | ⊐ | ⌣ | ⊐ | ⌣ | ⎮ | ⌣ | # |
| ⋀ | ⋀ | ⌣ | ⌣ | ⋀ | ⎮ | ⌣ | # |
| ⎮ | ⎮ | ⎮ | ⎮ | ⎮ | ⎮ | ⎮ | # |
| ⌣ | ⌣ | ⌣ | ⌣ | ⌣ | ⎮ | ⌣ | # |
| # | # | # | # | # | # | # | # |

Fig. 3: Entailment Composition Rules

*Categorical value domain match*: Given a pair of attributes $(a_i, a_j)$ with *categorical* value extents, $([\![a_i]\!], [\![a_j]\!])$, we define $d^v([\![a_i]\!], [\![a_j]\!])$, the categorical value domain match, as a multi–valued feature. The value of $d^v$ is determined based on the value domains of $([\![a_i]\!]$ and $[\![a_j]\!])$. A domain $\mathtt{d}_i^j$ of attribute $a_j$ from a table $S_i$ is a collection of representative tokens shared with $[\![a_j]\!])$. As such, given another domain $\mathtt{d}_k^i$ of an attribute $a_i$ from some table $S_k$, equality between $\mathtt{d}_i^j$ and $\mathtt{d}_k^i$ leads to $EQ$, containment leads to $FWD$ or $REV$, partial overlap leads to $COV$, and non–overlap (i.e, different domains) leads to $IND$. In practice however, $FWD$ or $REV$ are not possible because the domain identification process uses *D4* which aims for a minimal domain identification [18].

*Numerical value domain match*: Given a pair of attributes $(a_i, a_j)$ with *numerical* value extents, $([\![a_i]\!], [\![a_j]\!])$, we define $d^k([\![a_i]\!], [\![a_j]\!])$, the numerical value domain match, as a binary feature (i.e., $EQ$ or $IND$) between $([\![a_i]\!]$ and $[\![a_j]\!])$. We ground $d^k$ in the Kolmogorov–Smirnov (KS) statistic [6] that allows us to evaluate whether the two corresponding extents, seen as samples, are drawn from the same distribution (i.e., domain).

Given a target table $H \in \mathcal{D}$, similarly to the intensional case, we perform pair–wise processing of attribute extent pairs, as illustrated in the bottom part of Figure 2 and described next.

**Extensional Domain Extraction.** In analyzing extensional information, we start from the assumption that in order to produce an entailment relation between two attributes they need to be $q^v$–equivalent. Once that happens, we use the results of a previous run of *D4* at data lake–level to obtain the domain of each attribute and to extract $d^v$ for categorical attribute pairs and $d^k$ for numerical attribute pairs. D4 leverages value co-occurrence information across columns in a dataset to output a set of domains discovered by gathering contextual information for terms within columns in a set of tables. Full details of how *D4* achieves domain discovery are available in [18].

Once the extensional features are extracted for a given attribute pair, we employ Algorithm 3 to infer a composite extensional entailment. In the algorithm, we first obtain the domains for each attribute of the input pair (i.e., Lines 4 and 5). Equivalent domains lead to $EQ$, while attributes with different domains are further processed using GETCONTAINMENTREL, which takes two domains as arguments and applies the rules defined in Table 2 to infer a semantic relation between the given attributes.

---

**Algorithm 3** Compute extensional entailment for categorical attributes

---

 1: *Input:* categorical attributes $x$, $y$
 2: *Output:* $e_{\mathrm{d}}$ entailment relation between categorical attributes
 3: **function** GETDOMAINRELATION($x$, $y$)
 4:     $d_x \leftarrow x.getDomain()$
 5:     $d_y \leftarrow y.getDomain()$
 6:     **if** !isSameDomain($d_x$, $d_y$) **then**
 7:         $DT_x \leftarrow d_x.getDomainTerms()$
 8:         $DT_y \leftarrow d_y.getDomainTerms()$
 9:         $e_{\mathrm{d}} \leftarrow$ GETCONTAINMENTREL($DT_x$, $DT_y$)
10:     **else**
11:         $e_{\mathrm{d}} \leftarrow equivalence$
12:     **end if**
13:     **return** $e_{\mathrm{d}}$
14: **end function**

---

| $e_{\mathbf{domain}}(\mathbf{a,b})$ | Semantic Relation |
|---|---|
| $D_a = D_b$ | $\equiv$ |
| $D_a \subset D_b$ | $\sqsubset$ |
| $D_a \supset D_b$ | $\sqsupset$ |
| $D_a \cap D_b$ | $\smile$ |
| $D_a \neq D_b$ | $\#$ |

Table 2: Domain Containment Relation.

Finally, for the numerical case, we only consider $EQ$ and $IND$ as the possible values of $d^k$ and heuristically choose between them based on a $d^k_{threshold}$ threshold (i.e., a KS–statistic $> d^k_{threshold}$ results in $EQ$, and in $IND$ otherwise). $d^k_{threshold}$ is obtained through Equation 1, a common threshold used with the KS statistic, i.e., the 95% critical value of the KS statistic [6].

$$d^k_{threshold} = 1.36\sqrt{\frac{1}{|[\![a_i]\!]|} + \frac{1}{|[\![a_j]\!]|}} \tag{1}$$

for a pair of numerical value extents $([\![a_i]\!], [\![a_j]\!])$.

### 3.4 Entailment composition

We have described how, given a pair of attributes $(a_i, a_j)$ we extract the NLI–specific value for $[l^f, l^h, s^h, t^h]$ at intensional level, and for $[q^v, d^v, k^v]$ at extensional level. Additionally, we have described how each of the two feature collections is aggregated to a single relation type using Algorithm 2 and Algorithm 3, respectively. We now describe how the two types of relations, *viz.* intensional and extensional, are combined: attributes that are extensionally independent are by default labelled with $IND$, while extensionally related attributes are labelled by their intensional features. The full set of composition rules is defined in Figure 4.

| Extensional \ Intensional | $\equiv$ | $\sqsubset$ | $\sqsupset$ | $\wedge$ | $\mid$ | $\smile$ | # |
|---|---|---|---|---|---|---|---|
| $\equiv$ | $\equiv$ | $\sqsubset$ | $\sqsupset$ | $\wedge$ | $\mid$ | $\smile$ | # |
| $\sqsubset$ | $\equiv$ | $\sqsubset$ | $\sqsupset$ | $\wedge$ | $\mid$ | $\smile$ | # |
| $\sqsupset$ | $\equiv$ | $\sqsubset$ | $\sqsupset$ | $\wedge$ | $\mid$ | $\smile$ | # |
| $\smile$ | $\equiv$ | $\sqsubset$ | $\sqsupset$ | $\wedge$ | $\mid$ | $\smile$ | # |
| # | # | # | # | # | # | # | # |

Fig. 4: Entailment Composition Rules

### 3.5 Transformer-based RNLI

Given a pair of attributes between which exists a similarity relationship, the RNLI approach described above assigns an explainability dimension to an existing, general–level similarity. Having described how we perform unsupervised NLP–based analysis for similarity explainability, we now discuss a potential alternative for explaining how the attributes of retrieved datasets relate. We construe this explainability task as a fine–tuning BERT step where we rely on a pre–train BERT model to generate similarity–preserving representations for intensional information (i.e., table and attribute names). With these representations in hand, we build a simple downstream classification model that takes as input pairs of attribute intensional representations and labels them with one of the NLI semantic relations from Table 1.

Note that, for this simple alternative approach, we only consider intensional information. This, in turn, requires the assumption that the analyzed data lake presents datasets with semantically–meaningful table and attribute names. We argue that such scenarios are common in practice. For example, most of the datasets available on the public data lake `www.data.gov.uk` present such table and attribute names. We leave the more complex task of performing transformer–based extensional analysis for future work.

### 3.6 Intensional attribute representations

At its core, BERT is a language representation model capable of generating context–aware embeddings (as opposed to the limited context awareness of Word2Vec [16] models) for words and sentences. Given a name $\mathtt{n}_i^S$ of some table $S_i$ (e.g., *University Rankings, Graduate Employability*, etc. from Figure 1) and an attribute name $\mathtt{n}_i^j$ (e.g., *Name, Location*, etc. from Figure 1), we construe each concatenation $\mathtt{n}_i^S||\mathtt{n}_i^j$ as a sentence and represent it in an embedding space $\mathbb{R}^d$, offered by the pre–trained BERT model. We, thus, leverage the semantic awareness property of BERT to represent *sentences* and use such representation in a downstream classification task to identify specific semantic similarity types, as we now describe.

### 3.7 Supervised NLI labelling

Given a pair of *table name*$||$*attribute name* concatenations, for which we already have identified signals of general similarity (e.g., at intensional or extensional levels using

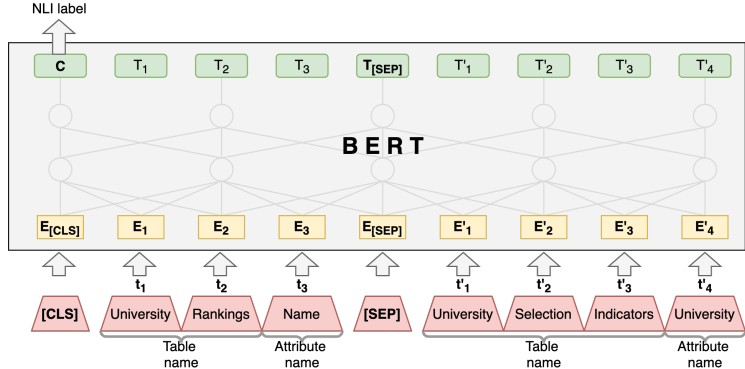

Fig. 5: BERT for NLI labeling. Pre–training is used to generate intensional attribute representations. Fine–tuning is used to label pairs of representations with NLI labels.

approaches such as $D^3L$ [2]), we use the model pictured in Figure 5 with a fine–tuning BERT task to further explain the type of similarity existing between the two attributes.

Specifically, and with reference to Figure 1, given the pair (*University Rankings Name, University Selection Indicators University*), corresponding to the first attributes from source $S_1$ and target $H$ in Figure 1, respectively, we feed it to a BERT neural network. During training, we also feed the pair's corresponding NLI label (i.e., from Table 1). Using its pre–trained weights, the model firstly generates a semantics–aware attention–based representation for each of the input sentences and optimizes new weights for a classifier whose aim is to label the pair appropriately.

## 4  Evaluation

We firstly evaluate the quality of our two methods for identifying inter–column entailment. We then compare the results against a similarity discovery technique from the state–of–the–art, $D3L$ [2]. Thirdly, we evaluate the quality of our proposed model when using it to explain the relationship between known similar columns. Finally, we perform an ablation analysis to determine how extensional and intensional features contribute to the explanations.

**Evaluation data:** The models are evaluated on ∼600 tables from real–world UK open government data[4], with information from seven domains, such as business, education, salaries, public service, etc.. The same dataset in used in [2].

**Experimental setup:** To parse and capture linguistic features on dataset elements we used the Stanford Core NLP library [15]. Additionally, we employ Wordnet [17] as an external lexical source to capture semantic features such as synonymic and taxonomic representations of attribute descriptors. For the transformer–based approach, we used a BERT pre–trained model from the Python *transformers* package[5]. During fine–tuning,

---

[4] www.data.gov.uk

[5] https://pypi.org/project/transformers/

|       | Precision |      | Recall |      | F1-Score |      |
|-------|-----------|------|--------|------|----------|------|
| Class | RNLI | BERT | RNLI | BERT | RNLI | BERT |
| IND   | 0.80 | 1.00 | 0.99 | 1.00 | 0.88 | 1.00 |
| ALT   | 0.91 | 0.81 | 0.73 | 1.00 | 0.81 | 0.90 |
| EQ    | 0.83 | 1.00 | 0.54 | 0.63 | 0.66 | 0.77 |
| REV   | 0.56 | 1.00 | 0.55 | 0.88 | 0.55 | 0.94 |
| FWD   | 0.72 | 1.00 | 0.87 | 0.57 | 0.79 | 0.73 |

Table 3: RNLI & BERT results

we trained the model using the Adam optimizer [11] with a linearly–increasing learning rate starting from $10^{-5}$, for *10* epochs and with a batch sized of *5* [6]. We trained the model on $70\%$ of the data, validate it (i.e., hyperparameter optimization) on $10\%$ of the data, and tested it on the remaining $20\%$. All data, source code, baselines and hyperparameter tuning settings are shared[7] for reproducibility purposes.

**Baselines and reported measures:** We evaluate both the composite entailment and the transformer–based models in terms of precision, recall and F1 scores. Using the same measures, we also perform a comparative evaluation between our RNLI proposals and the $D^3L$ similarity framework [2]. For the purpose of computing the evaluation metrics, we use the same ground truth used in [2], where each attribute pair is associated with a binary relatedness representation (i.e., related/unrelated). Out of the approx. *600,000* attribute pairs recorded in the ground truth, we explained *13,000* pairs by labeling each such pair with one of our entailment relationship types from Table 1. Lastly, we use the *600,000* attribute pairs for performing the comparative evaluation against $D^3L$ and the more explicit *13,000* attribute pairs for evaluating the explainability potential of our proposal.

### 4.1   Inference performance

Both composite entailment and the transformer–based models are initially evaluated in terms of per–class precision, recall and F1 score. The hypothesis in this experiment is that *our intensional and extensional analysis can indicate not only similar attribute pairs, but also explain the semantic relation type.* Additionally, we hypothesize that, when there is sufficient exemplar data available, *the semantic relation identification can effectively be construed as a BERT fine-tuning task.* We report the per–relationship type results in Table 3.

Overall, the supervised transformer–based approached proves superior to the unsupervised method. However this is conditioned by the existence of labeled training data. In this experiment, approximately $13,000$ attribute pairs had associated NLI labels.

Both RNLI and BERT tend to misclassify related attribute pairs as equivalent. This leads to poorer precision and recall for $EQ$. In the case of $REV$ and $FWD$ the poor recall of both RNLI and BERT can be explained by the relatively small number of

---

[6] These parameters lead to the best results during validation.

[7] https://bit.ly/3lrb5JD

| Domain | Accuracy | | Precision | | Recall | | F1-Score | | $\Delta$ F1-score |
|---|---|---|---|---|---|---|---|---|---|
| | D3L | RNLI | D3L | RNLI | D3L | RNLI | D3L | RNLI | |
| Business | 0.936 | 0.952 | 0.949 | 0.930 | 0.855 | 0.927 | 0.900 | 0.929 | **0.029** |
| Schools | 0.898 | 0.933 | 0.603 | 0.726 | 0.660 | 0.783 | 0.630 | 0.753 | **0.123** |
| Elections | 0.771 | 0.779 | 0.883 | 0.905 | 0.216 | 0.241 | 0.346 | 0.380 | **0.034** |
| Flights | 0.925 | 0.937 | 0.660 | 0.790 | 0.224 | 0.333 | 0.335 | 0.469 | **0.134** |
| Food | 0.916 | 0.927 | 1.000 | 0.912 | 0.317 | 0.454 | 0.482 | 0.606 | **0.124** |
| Public Spending | 0.764 | 0.915 | 0.999 | 1.000 | 0.540 | 0.833 | 0.701 | 0.909 | **0.208** |
| Salaries | 0.861 | 0.774 | 1.000 | 1.000 | 0.803 | 0.678 | 0.891 | 0.808 | $-0.083$ |
| **Average** | 0.867 | 0.888 | 0.871 | 0.895 | 0.516 | 0.607 | 0.612 | 0.694 | **0.082** |

Table 4: D3L & RNLI results

instances with this label in our test data (i.e., less than 20 in each case). Thus, a single miss can have a significant impact on the recall results.

### 4.2 Comparative analysis

In this experiment, we aim to compare the performance of our unsupervised proposal and a similarity–focused baseline, i.e., $D^3L$ [2]. For this purpose we:

1. Transform our multi–class problem to a binary class problem so that the comparison is possible. The mapping that enables this transformation is: $IND \rightarrow dissimilar$; $\{EQ, ALT, REV, FWD\} \rightarrow similar$.
2. Evaluate the two approaches on the entire $D^3L$ ground truth, i.e., *600,000* attribute pairs. Consequently, we do not include the transformer–based approach in this evaluation.
3. Since $D^3L$ is a ranked–retrieval approach, i.e., it retrieves the top–$k$ most similar attributes to a given query attribute, for the purpose of comparison, we randomly pick 20 tables and use their columns as the queries for both the compared approaches. For $D^3L$, $k$ is set in accordance to the size of each domain existing in the data and present in Table 4. For example, there are approx. 50 tables with *Salary* information and, therefore, for each run of $D^3L$ with a target attribute from the same domain $k = 50$. The purpose of this setting is to avoid penalizing $D^3L$'s recall by setting a fixed, potentially too small, value of $k$.

Table 4 shows the per–domain (i.e., there are seven different domains in the evaluation dataset) accuracy, precision, recall, and F1–score values, and their average. Overall, it can be concluded that the RNLI approach can be reliably converted to a similarity–focused approach. When this happens, RNLI performs better in most cases. The exception, *Salaries*, is due to a high concentration of numerical attributes that is specially addressed in $D^3L$.

### 4.3 Inference explanations

Previous experiments prove the comparable performance of RNLI to the similarity discovery state–of–the–art. In this experiment, we aim to confirm similar levels of

| Relation | Precision | Recall | F1 Score |
|----------|-----------|--------|----------|
| EQ | 0.866 | 0.742 | 0.799 |
| REV | 0.676 | 0.556 | 0.610 |
| FWD | 0.545 | 0.667 | 0.600 |
| ALT | 0.944 | 0.939 | 0.942 |
| IND | 0.904 | 1.000 | 0.950 |

Table 5: RNLI Composite Entailment Relation Performance

performance, this time with a focus on RNLI's similarity explainability potential. To this end, we consider a scenario in which $D^3L$ is initially employed to efficiently identify similar attributes in a data lake. Then, we employ RNLI as a semantic explainability mechanism to further explain the relationships between the attribute pairs that have been deemed related by $D^3L$. The results are shown in Table 5.

Overall, RNLI proves reliable in explaining similarity relations grounded on $EQ$, $ALT$, and $IND$. As before, the results are moderate for $REV$ and $FWD$, mostly due to a relatively small number of instances with these labels in our test data.

### 4.4   Ablation Study

To have a better understanding of how extensional and intensional features contribute to the final inference, an ablation study is performed in this experiment. To this end, we first isolate intensional features and compute semantic relations. In a second run, we focus exclusively on extensional features to compute the results. Finally we show the results for the combined computation with all feature types considered. Figure 6 shows the obtained results.

Intensional information proves superior to the extensional evidence in identifying NLI relations. In the cases of $EQ$ and $IND$, this is because, most of the time, the tables and attributes in our test data lake present semantically meaningful names. Reasoning based on these names is more reliable than inferring NLI relations from value domain analysis. In the cases of $REV$, $FWD$ and $ALT$, extensional results are close to zero because *D4* aims for a close to minimal and disjoint domain identification. This means that RNLI will mostly infer $EQ$ or $IND$ when analyzing *D4* domains. Lastly, in line with our core desideratum in this paper, the combination of intensional and extensional evidence leads to the strongest RNLI results.

## 5   Conclusions

This paper presented the Relational Natural Language Inference Model, a composite entailment framework that models the data lake data discovery problem through natural language inference. We empirically demonstrated an end-to-end process to compute semantic relations at different levels of abstraction by leveraging different sources of signal present in data sources. Additionally, this study was carried out by taking consideration of the explainability of the candidate source relations. Overall, the RNLI

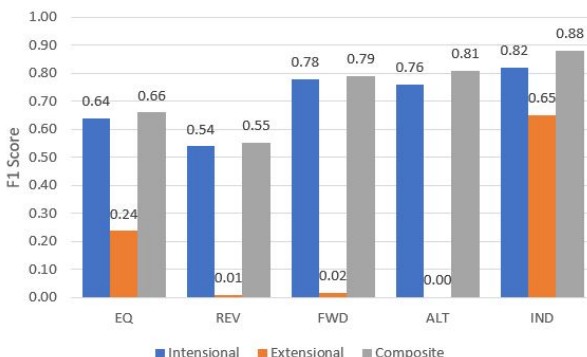

Fig. 6: Ablation Study

model outperforms existing approaches, improving the way inferences are computed and represented in an explainable format. This contributes to a better understanding and semantic control of the inference process. Finally, we see our approach as a mechanism for providing interpretable semantic relations for integration tasks, such as schema matching and entity resolution.

**Acknowledgements.** Mario Ramirez is supported by the Mexican National Council for Science and Technology (CONACYT). Alex Bogatu is supported by Innovate UK.

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
