# OpenReview forum: "Natural Language Inference over Tables: Enabling Explainable Data Exploration on Data Lakes"
_eswc-conferences.org/ESWC/2021/Conference/Research_Track — ESWC 2021 Research_

### Official Review · AnonReviewer5 · 2021-01-12
**RNLI review**

**Rating:** 1
**Confidence:** 5
**Impact:** 3
**Design And Technical Quality:** 3

**Review:**

This paper describes an approach exploiting basic natural language processing (POS tags, noun phrase detection, modifier detection on table names and attributes - the paper also considers so-called extensional attributes such as the types of various attributes and their value domains) to creating a set of natural language relations between structured tables in a data lake such as data.gov.  The idea is definitely interesting - and the paper makes the novelty of the approach reasonably clear.  The approach also tries to use this natural language relations later to train a BERT based model on a supervised task to infer such relations from names of tables and attributes.  An evaluation is performed comparing supervised and unsupervised models on a baseline from prior work (although that work seems to only classify attribute pairs as similar or dissimilar).   Also interesting is the evaluation on the unsupervised model and a prior system D3L.

**Anonymity:**

Yes, I would like my review to remain anonymous.

**Reuse And Availability:**

3: Medium

**Strong Points:**

1.  The paper attacks an important problem and to my knowledge is novel - applying detailed NLP processing to specify semantic relations between columns is a nice idea, as is training NLP language models to perform this classification.
2.  The evaluation section is strong, and has a nice set of studies including ablation studies that show better performance on intensional attributes than extensional attributes.

**Subreviewer:**

I submitted this review.

**Weak Points:**

1.  The writing and in particular the motivation for each of the relations and how they contributed to the different relations needs to be improved - in its current state, the different relations and their compositions seem to be presented in a disorganized fashion.  For instance, Table 1 was a nice description of all the relation types.  However, instead of using Table 1 to define how each of those relations were created (and how they were composed if the table and attribute did not agree), Section 3.2 broke it out into a process by which entailment relations were observed - which made a reader wonder for instance how something got to be a negation.  Also, for instance, many of the relations if I understood it correctly were only inferred intensionally.  Negation is never mentioned after table 1.  It appears from the text that cov, and even forward and backwards entailment could not be inferred extensionally -- see the following statement: "As such, given another domain d of an attribute ai from some table Sk, equality between dji and dik leads to EQ, containment leads to FWD or REV , partial overlap leads to COV , and non–overlap (i.e, different domains) leads to IND. In practice however, FWD or REV are not possible because the domain
identification process uses D4 which aims for a minimal domain identification [18]".
2.  Also confusing was the composition.  According to Algorithm 2, for instance Regional University Rank - September is an alternation of University Rank - October.  Bur clearly regional ANYTHING could be covered by a more general ANYTHING.
3.  The paper needs to be self contained.  The use of D4 heavily to create the actual entailments requires a description that makes the paper self contained, so the reader can understand why COV/FWD/REV are not plausible.  Also the evaluation rests heavily on D3L - needs to be described in sufficient detail so one can understand what the actual dataset is about.
4.  If accepted it would be useful to make the BERT model available for evaluation by the scientific community, not just the code.

---

> ### Author Rebuttal · Authors · 2021-01-30
>
> Here we provide some responses to the reviewer's comments:
>
> -  "...in its current state, the different relations and their compositions seem to be presented in a disorganized fashion. For instance, Table 1 was a nice description of all the relation types. However, instead of using Table 1 to define how each of those relations were created (and how they were composed if the table and attribute did not agree), Section 3.2 broke it out into a process by which entailment relations were observed - which made a reader wonder for instance how something got to be a negation."
> >*Indeed, for writing space purposes we omitted this detail as it is explained in the references related to Natural Language Inference. In summary, negation is the relation that expresses exhaustive semantic exclusion in Natural Language Inference. In set theoretic definitions, negation implies non-overlapping sets which together exhaustively cover a given universe. Given this highly specific setting, such conditions rarely occur in the applicable space of our approach, as we do not know the universe. If we had this for some attributes, we could include negation. We are happy to clarify this in the paper improving content organisation.
> This comment has been used to discuss similar observations from Reviewer 3.*
> - "Also confusing was the composition. According to Algorithm 2, for instance Regional University Rank - September is an alternation of University Rank - October. Bur clearly regional ANYTHING could be covered by a more general ANYTHING."
> >*Algorithm 1 describes the process to compute entailments between noun phrases. In order to determine an entailment, when comparing noun phrases, a compositional ordering from head to modifier is established and used. Algorithm 2 is focused on how table and attribute information is composed together to generate an intensional entailment relation. We plan to use this feedback to clarify the algorithms section for better understanding of the entailment process, as well as adding example entailments as supplementary material.
> This comment has been used to discuss similar observations from Reviewer 3.*
> - "The paper needs to be self contained. The use of D4 heavily to create the actual entailments requires a description that makes the paper self contained, so the reader can understand why COV/FWD/REV are not plausible."
> >*We acknowledge that the current description of the use of D4 is minimal; we would be happy to elaborate on the use of D4 to make explicit more details on the plausibility of certain entailment relations given the context of D4. This in order to make the paper self contained.*
> - "Also the evaluation rests heavily on D3L - needs to be described in sufficient detail so one can understand what the actual dataset is about"
> >*We acknowledge that the current description of D3L is quite minimal; we would be happy to expand the current coverage to make explicit more details, on the types of evidence used in its indexes, and how they are combined to produce a similarity measure for ranking.*
> - "it would be useful to make the BERT model available for evaluation by the scientific community, not just the code."
> >*For reproducibility purposes, we have made our implementation available here: https://colab.research.google.com/drive/1a6QNefWANYSWcLErhXNgE7rhOtm_VhEU?usp=sharing . The model uses the bert-base-uncased version from Huggingface’s transformers Python library (https://github.com/huggingface/transformers).
> This comment has been used to discuss similar observations from Reviewer 4.*

---

### Official Review · AnonReviewer2 · 2021-01-13
**RNLI for data exploration with unsupervised and supervised methods**

**Rating:** 1
**Confidence:** 5
**Impact:** 4
**Design And Technical Quality:** 4

**Review:**

The paper presents two methods, one based on linguistic and semantic entailment rules, and another supervised classification method based on BERT transformers. The goal of the paper is to find entailment relations between source and target tables. The unsupervised method uses intentional (schema) and extensional (data) evidence, which are eventually combined with entailment rules. The supervised method is only applied at intentional level.

The paper is well written and organized, with a good description of the methods and a proper evaluation of them. However, related work miss important references to similar tasks in the literature.

The unsupervised method presents some degree of novelty, although much of the techniques have been already proposed for schema matching and ontology alignment in the literature (which are omitted in this paper). The main novelty is in the entailment rules that allow combine different evidence sources.

The supervised method seems a first attempt to tackle this problem with novel NLP techniques. It has been limited to use BERT for classifying pairs of labelled pairs table/attribute based on a pre-trained BERT model. However BERT is able to make NLI directly so authors should test a fine-tuned system for inference instead of a classifier. Also there is no details nor references to the pre-trained model of BERT (base, large, cased, uncased ???)

As for the extensional part, there is relevant work that has been omitted in the paper. For example, the work of Jiaoyan Chen et al. "Learning Semantic Annotations for Tabular Data. IJCAI 2019: 2088-2094" presents a way to get embeddings for representing the extensional part of tables.

Finally, I'm not sure if "Data Lake" is a good scenario for the proposed tools. Experiments are carried out on Open Data, so the paper should include "Open Data" in the title instead of "Data Lake". A Data Lake have additional structures and catalogues that help the location of source tables. Additionally, authors can discuss the application of these methods to Linked Open Data as well.


**Anonymity:**

Yes, I would like my review to remain anonymous.

**Reuse And Availability:**

3: Medium

**Strong Points:**

Well-written paper
Interesting methods for finding table entailments
Good evaluation of the methods


**Subreviewer:**

I submitted this review.

**Weak Points:**

The paper focus on data lakes but the method seems more appropriate for Open Data
Missing relevant references (mainly within Semantic Web area)
Missing details in the supervised method

---

> ### Author Rebuttal · Authors · 2021-01-30
>
> Here we provide some responses to the reviewer's comments:
>
> - -"Related work misses important references to similar tasks in the literature[...]As for the extensional part, there is relevant work that has been omitted in the paper. For example, the work of Jiaoyan Chen et al. "Learning Semantic Annotations for Tabular Data. IJCAI 2019: 2088-2094" presents a way to get embeddings for representing the extensional part of tables."
> >*The related work section currently focuses on data discovery, to which we aim to contribute, and on language inference techniques, on which we build. Several reviewers would have liked us to consider other adjacent areas as well, in particular information integration, ontology alignment and semantic annotation. We are happy to extend the related work section to include these areas.
> We think that our work complements existing results in all these areas. In information integration, though we could provide an additional type of evidence to composite matchers, we note that in practice most data integration is highly manual, and thus our emphasis on explainability is relevant to users of data integration systems where users take fine-grained control over most decisions.  In relation to work on ontology alignment, we are trying to make minimal assumptions about the semantics available about the source and the target, while still inferring useful semantic relations between attributes. In relation to semantic annotation, in addition to the work of Jiaoyan Chen et al., there has been recent work evaluating the use of BERT as a supervised strategy (http://www.vldb.org/pvldb/vol13/p2549-li.pdf), to which we could easily relate our approach, though we note that our natural language inference angle seems to be distinctive.
> This comment has been used to discuss similar observations from Reviewer 1.*
> -  -"I'm not sure if 'Data Lake' is a good scenario for the proposed tools. Experiments are carried out on Open Data, so the paper should include 'Open Data' in the title instead of 'Data Lake'[...]The paper focus on data lakes but the method seems more appropriate for Open Data Missing relevant references (mainly within Semantic Web area)"
> >*Several reviewers have commented on our choice of data lakes as a motivating context. We are motivated by the rapid growth in data lakes, and associated growth in data catalogs to help users to come to terms with the available data (the Data Catalog market is worth several hundred million dollars and growing rapidly: https://www.marketsandmarkets.com/Market-Reports/data-catalog-market-48918216.html).  However, we agree with the reviewers that the techniques developed are not specific to data lakes, and are equally relevant to other sources of autonomously produced data. This is reflected in the evaluation, which makes use of open government data. We would be happy to emphasise this point in the paper, and even to change the title if the emphasis on data lakes is felt to be misleading.
> This comment has been used to discuss similar observations from Reviewer 1.*
>
> - "Missing details in the supervised method"
> >*We have not provided full details on the supervised approach since, in practice, it can be conducted using a BERT fine-tuning task. The details of such a task has been extensively described in the original BERT paper by Devlin et al. (https://arxiv.org/abs/1810.04805 ). For reproducibility purposes, we have made our implementation available here: https://colab.research.google.com/drive/1a6QNefWANYSWcLErhXNgE7rhOtm_VhEU?usp=sharing . The model uses the bert-base-uncased version from Huggingface’s transformers Python library (https://github.com/huggingface/transformers).
> This comment has been used to discuss similar observations from reviewer 3.*
>
> - "BERT is able to make NLI directly so authors should test a fine-tuned system for inference instead of a classifier."
> >*BERT is indeed capable of NLI. However, the NLI task we consider in our paper is focused on tabular data and it is defined by the specific types of relationships we aim to infer, as described in the paper. Thus, the task of NLI defined in such a way can be approached similarly to a classification problem. Similar classification--oriented approaches have been proposed before for evaluating BERT for NLI, e.g., N. Jiang et al., Evaluating BERT for natural language inference: A case study on the CommintmentBank, EMNLP 2019.*
>
> - "Also there are no details nor references to the pre-trained model of BERT (base, large, cased, uncased ???)"
> >*For reproducibility purposes, we have made our implementation available here: https://colab.research.google.com/drive/1a6QNefWANYSWcLErhXNgE7rhOtm_VhEU?usp=sharing . The model uses the bert-base-uncased version from Huggingface’s transformers Python library (https://github.com/huggingface/transformers).*

---

### Official Review · AnonReviewer3 · 2021-01-14
**Review for A Natural Language Inference Framework for Data Exploration on Data Lakes**

**Rating:** 1
**Confidence:** 4
**Impact:** 3
**Design And Technical Quality:** 3

**Review:**

The paper describes the use of Natural Language Inference in data discovery to explain how attributes of discovered data sets relate to target ones. The authors propose both an unsupervised and supervised solution for deriving the semantic relations.

For a better redability, Table 1 should be moved from the introduction to section 3.

How the Negation semantic relation is derived is not described in the paper.

There are some interrogations that arrise concerning the proposed algorithms:
In Algorithm 1, it is unclear why is "not synonymic" leads to independance. Also the taxonomic relations will be calculated only if there is synonymy between the head tockens.
Why is it in Algorithm 2, that only IND, FWD and REV for table names are considered. How about the combination of table and attribute names equivalence with attribute name like it is the case for "Graduate Employability"."Rank" and H."Employability Rank". It seems that a comparison of different levels of entailment is missing.

Fig.3 should be titled "Entailement Projection Rules"

The transformer-based RNLI approach would have deserved more attention, it is relatively shortly described. Also its evaluation is in my opinion insuficiant. Being trained on 70% and only tested on 10% of the whole dataset, besides the fact that the whole has been operated on very similar data.

It is unclear how the ground truth has been created. This should be clarified. Also it could be interesting to consider a comparative analysis on a larger set considering an additional tool.


**Anonymity:**

Yes, I would like my review to remain anonymous.

**Reuse And Availability:**

2: Low

**Strong Points:**

* The use of NLI for data discovery is innovative.
* The notion of adding explainability via semantic relations is very interesting.
* The combination of both intensional and extensional evidence types to derive similarity relationships.


**Subreviewer:**

I submitted this review.

**Weak Points:**

* There is an issue with the completeness of the algorithms, they seem not to cover all possible combinations as mentioned above.
 * The supervised approach is succinctly described with insufficient evaluation.
* An explanation of how the ground truth for the evaluation has been created is missing especially considering the size of the attributes pairs set (600K). Also an additional tool for a comparative study would add more significance to the results.

---

> ### Author Rebuttal · Authors · 2021-01-29
>
> Here we provide some responses to the reviewer's comments:
>
> - "How the Negation semantic relation is derived is not described in the paper."
> >*Indeed, for writing space purposes we omitted this detail as it is explained in the references related to Natural Language Inference. In summary, negation is the relation that expresses exhaustive semantic exclusion in Natural Language Inference. In set theoretic definitions, negation implies non-overlapping sets which together exhaustively cover a given universe. Given this highly specific setting, such conditions rarely occur in the applicable space of our approach, as we do not know the universe. If we had this for some attributes, we could include negation. We are happy to clarify this in the paper.*
>
> - "In Algorithm 1, it is unclear why "not synonymic" leads to independence. Also the taxonomic relations will be calculated only if there is synonymy between the head tokens."
> >*Algorithm 1 describes the process to compute entailments between noun phrases. In order to determine an entailment, when comparing noun phrases, a compositional ordering from head to modifier is established and used. Algorithm 2 is focused on how table and attribute information is composed together to generate an intensional entailment relation. We plan to use this feedback to clarify the algorithms section for better understanding of the entailment process, as well as adding example entailments as supplementary material.*
>
> - "The supervised approach is succinctly described with insufficient evaluation."
> >*We have not provided full details on the supervised approach since, in practice, it can be conducted using a BERT fine-tuning task. The details of such a task has been extensively described in the original BERT paper by Devlin et al. (https://arxiv.org/abs/1810.04805 ). For reproducibility purposes, we have made our implementation available here: https://colab.research.google.com/drive/1a6QNefWANYSWcLErhXNgE7rhOtm_VhEU?usp=sharing . The model uses the bert-base-uncased version from Huggingface’s transformers Python library (https://github.com/huggingface/transformers).*
>
> - "The transformer-based RNLI approach would have deserved more attention, it is relatively shortly described. Also its evaluation is in my opinion insuficient. Being trained on 70% and only tested on 10% of the whole dataset, besides the fact that the whole has been operated on very similar data."
> >*We have not provided full details on the supervised approach since, in practice, it can be conducted using a BERT fine-tuning task. The details of such a task has been extensively described in the original BERT paper by Devlin et al. (https://arxiv.org/abs/1810.04805 ). For reproducibility purposes, we have made our implementation available here: https://colab.research.google.com/drive/1a6QNefWANYSWcLErhXNgE7rhOtm_VhEU?usp=sharing . The model uses the bert-base-uncased version from Huggingface’s transformers Python library (https://github.com/huggingface/transformers).
> With respect to the training/test/validation sizes, there is a typo in the paper when we explain the BERT experimental setup (page 12). We used 70% of the data for training, 10% for validation, and 20% for testing, not 15% mentioned in the paper now. Validation is used for parameter tuning and/or  for early stopping of the training process. We will update the paper with the correct information.*
>
> - "It is unclear how the ground truth has been created. This should be clarified. Also it could be interesting to consider a comparative analysis on a larger set considering an additional tool."
> >*The current ground truth has been manually and, when possible,  programmatically created by a human annotator. Specifically, a few hundred attribute pairs have initially been manually annotated and their relationships have been formalized in custom scripts that have then been used to identify additional pairs with similar relationships between their members.
> 	Additional tools from the area of data discovery that could be considered in the evaluation include R.C. Fernandez et al. Aurum: A Data Discovery System, ICDE 2018 and F. Nargesian et al. Table Union Search on Open Data, PVLDB 2018. Similar to D3L, none of these offer an explainability dimension and their performance inferiority to D3L has already been shown in the D3L paper (A. Bogatu et al. Dataset Discovery in Data Lakes, ICDE 2020) on the same dataset used in our experiments.*

---

> > ### Comment · AnonReviewer3 · 2021-02-03
> > **Reconsidering rating after rebuttal**
> >
> > Thanks to the detailed answer and explanations, I am willing to reconsider my score a bit higher. This under the condition that the authors take into account the review comments and update the paper accordingly as mentioned in their rebuttal.

---

### Official Review · AnonReviewer1 · 2021-01-14
**Review: A Natural Language Inference Framework for Data Exploration on Data Lakes**

**Rating:** 1
**Confidence:** 4
**Impact:** 3
**Design And Technical Quality:** 2

**Review:**

The authors address the problem of finding relevant data sets in a data lake for a given task. The target dataset is defined as a set of columns. Natural Language Inference (NLI) is used to calculate the semantic relations, by obtaining several types of evidence and combining them. Two approaches for inferring the semantic relations are proposed, the first approach using intensional and extensional evidence to provide explainable results, the second uses BERT to build a classification model.

The authors contribute to data discovery by building on language inference. The state of the art in other areas, e.g. information integration, is not considered. The paper contributes to data discovery providing semantic relations that specify how the discovered data is relevant for the target data. After relevant data has been found, the classification of the semantic relations seems a task better suited for other techniques in the area of information integration, which limits the potential impact of the paper.


**Comments:**

Section 2, Related Work, mentions that “ontology based approaches [...] are limited in scope by their dependence on external knowledge bases”. Ontologies, as an approach for web-scale knowledge bases, rely on consensus for alignment and interoperability. The scope is not limited by the web-scale suitability, but extended by it.

The running example considers several data sets that are relevant for a particular target table, built apparently by joining the relevant data sets. This is rarely the case. Neither the examples or the explanations make clear how more complex scenarios would be handled, e.g.

1. How attributes population and GDP would be considered relevant for the target of GDP per capita.
1. How the algorithms manage mereologies, possibly more usual in data lakes than taxonomies. $YouthUnemploymentRate \sqsubset UnemploymentRate$ ?
1. How the extensional values of an attribute could be related to target attributes (pivot).

Minor comment: Table 4 should read $D^3L$ but reads $D3L$.


**Anonymity:**

Yes, I would like my review to remain anonymous.

**Reuse And Availability:**

2: Low

**Strong Points:**

The paper focuses on data lakes, however the techniques may be inspiring for approaches working over any arbitrary set of data sets, e.g. mixing open data with the data lake of some organisation. Such an approach would open many possibilities for future lines (not discussed in the paper), and challenges. For example the mapping between metadata and semantics may vary between organisations.

The extent to which the current approach relies on an homogeneous mapping (same language, same terms, same context,...) as may be found in a single data lake would limit its applicability to merging data from several sources.


**Subreviewer:**

I submitted this review.

**Weak Points:**

The two main weak points of the paper are:
1. Not considering other areas that may be relevant for the problems addressed in the paper, e.g. information integration.
1. Not considering downstreaming tasks and what kind of results would be most helpful, e.g. data analytics.

A few sentences may be enough to explain why they are not relevant, if that is the case.

The authors propose an approach to find the semantic relations between data sets, but _the approach does not include the integration of the datasets_ into a single dataset, i.e. the output is generated for human consumption. The possibility of using the semantic relations as an input for another algorithm is not discussed. This is normally one of the promises of semantic technologies, and would allow the integration of the proposed algorithm in automated DataOps pipelines.

The evaluation uses data from www.data.gov.uk. Using several sources should provide a better evaluation, especially covering the casuistry in heterogeneous data lakes. It is unclear to what extent the findings with data.gov.uk would generalise to data lakes in general. For example, most data lakes are not public.

The “Motivating Scenario” motivates a solution, but not necessarily the proposal in this paper. In particular, the motivation or rationale behind the choice of the semantic relations might be interesting, for example when considering how it informs downstreaming tasks, e.g. integration of the data sets.

A proper characterization of the problem should allow filtering or ranking the approaches for it. Such characterization should include the objective. For instance, in the construction of the dataset, precision may be more relevant than recall. Errors in precision may introduce incorrect data, leading to wrong conclusions; while errors in recall may miss redundant (overlapping) data, with no further consequence.

The impact is limited, as the problem solved might be better addressed with techniques not considered.

Reusability and availability could be improved, with open source and a better study of the use cases that would be covered.

---

> ### Author Rebuttal · Authors · 2021-01-29
>
> Here we provide some responses to the reviewer's comments:
>
> - "The state of the art in other areas, e.g. information integration, is not considered."
> >*The related work section currently focuses on data discovery, to which we aim to contribute, and on language inference techniques, on which we build. Several reviewers would have liked us to consider other adjacent areas as well, in particular information integration, ontology alignment and semantic annotation. We are happy to extend the related work section to include these areas.
> We think that our work complements existing results in all these areas. In information integration, though we could provide an additional type of evidence to composite matchers, we note that in practice most data integration is highly manual, and thus our emphasis on explainability is relevant to users of data integration systems where users take fine-grained control over most decisions.  In relation to semantic annotation, in addition to the work of Jiaoyan Chen et al., there has been recent work evaluating the use of BERT as a supervised strategy (http://www.vldb.org/pvldb/vol13/p2549-li.pdf), to which we could easily relate our approach, though we note that our natural language inference angle seems to be distinctive.
> This comment has been used to discuss similar observations from Reviewer 1.*
>
> - "Ontologies, as an approach for web-scale knowledge bases, rely on consensus for alignment and interoperability. The scope is not limited by the web-scale suitability, but extended by it."
> >*We agree that the present choice of words may not accurately reflect our intended argument. In the statement "However, these approaches are limited in scope by their dependence on external knowledge bases.” we refer to methods, such as [12], that rely on different types of external information, e.g., knowledge bases,  to derive a feature extraction framework to be used with probabilistic and machine learning algorithms for table annotations. Such proposals assume high levels of notion-overlap between the used external knowledge bases and the input data. However, table data is often domain-specific and its entities may be insufficiently covered by web knowledge bases. We offer to make adjustments to the Related Work section to better reflect our intended argument.*
>
> - "Neither the examples or the explanations make clear how more complex scenarios would be handled..."
> >*GDP per capita: For this example the algorithm would not be able to align population with GDP per capita, only with GDP. This points into a quite relevant direction for future work. On the mereology: the proposed method focuses on mereological-type relations which are expressed by compositional patterns in complex nominals (YouthUnemploymentRate, UnemploymentRate).  We found this category to be highly representative when aligning attribute descriptors.*
>
> - -"Not considering downstreaming tasks and what kind of results would be most helpful, e.g. data analytics."
> >*It is fair to observe that the paper has not discussed alternative ways in which the inferred semantic relations could be used, as our emphasis was on explaining relationships between source and target attributes to users, e.g., in data catalogs. However, we acknowledge that the information obtained could be relevant to downstream algorithms, such as those for multi-criteria source selection or matching. We have been asked to expand the scope of the related work, and we would be happy to include a discussion on how the inferred relationships may be able to be used to inform other data integration tasks.*
>
> - "It is unclear to what extent the findings with data.gov.uk would generalise to data lakes in general. For example, most data lakes are not public."
> >*Although the data used in our experiments originates from the same repository (data.gov.uk), it represents information from different sources and different domains, uploaded to the public repository by different organisations, without obeying any global schemata or global rules imposed by that repository. Also, by the internal organization of data.gov.uk, each of the seven domains used in the evaluation can be seen as a mini-data lake and doing so would still not change or affect the claims and arguments of the paper.*
>
> - "The “Motivating Scenario” motivates a solution, but not necessarily the proposal in this paper."
> >*The motivation scenario in Section 1 outlines the context of the problem and our proposed approach with our contributions. Indeed, a better explanation can be formulated regarding interpretable semantic relations that inform downstream tasks. We are happy to improve the section for clarity.*
>
> - " Reusability and availability could be improved, with open source and a better study of the use cases that would be covered."
> >*The source code for the experimental workflow will be published on github with an associated docker image. The dataset will be published under a creative commons license.*

---

> > ### Comment · AnonReviewer1 · 2021-02-07
> > **Reply**
> >
> > >  we note that in practice most data integration is highly manual, and thus our emphasis on explainability is relevant to users of data integration systems where users take fine-grained control over most decisions
> >
> > When comparing $a_i$ an automated approach that needs manual curation with $a_j$ an explainable approach that informs and possibly assists the manual approach. We may _arguably_ say that $a_j$ is a step back from the state of the art. The point of adding the "adjacent areas" is _showing_ how the proposed approach is a step beyond the current state of the art, and not a lateral step from an area $a_1$ into another $a_2$, detached from the state of the art in $a_2$.
> >
> > > we refer to methods, such as [12], that rely on different types of external information
> >
> > Methods that do not use external information may also be relevant. In a data lake context, each data set may have an explicit or inferred schema. Description logic (and therefore ontologies) may help to describe each one of them and/or the relations between their terms (concepts). Such an ad-hoc ontology may be used as a global schema for the integration of the datasets in the data lake, with or without external inforamation. IIRC, several similar proposals exist in the ontology-based data access area.
> >
> > > the proposed method focuses on mereological-type relations
> >
> > This should be made more clear.
> >
> > >  we would be happy to include a discussion on how the inferred relationships may be able to be used to inform other data integration tasks
> >
> > Much appreciated.
> >
> > > uploaded to the public repository by different organisations
> >
> > The results variance between organisations would be an interesting evaluation metric. Extensive evaluation is only possible after extensive application, though. Good luck.
> >
> > > We are happy to improve the section for clarity.
> >
> > Thank you. Solving the previous points (in particular, the first point) should solve this one
> >
> > > will be published on github with an associated docker image
> >
> > Greatly appreciated.

---

### Official Review · AnonReviewer4 · 2021-01-15
**Valuable paper but not providing strong results**

**Confidence:** 2
**Impact:** 2
**Design And Technical Quality:** 4

**Review:**


The paper addresses the problem of finding semantically related attributes in database tables and provide an explanation of the findings.

The work proposes two approaches: one based on using Natural Language Inference in this structured setting of relational data (RNLI); the other based on BERT.
In the first case the also use extensional (instance) data to find matchings, while for BERT only intensional information is used.

The experiment section make a comparative analysis with a similarity-based existing solution, showing that the RNLI is comparable (and slightly superior) to a previous approach.

Overall, the paper can be improved considering the work done in the field of databases.
E.g., why is the focus on data lakes and not databases? there are many work in the database area very related to this problem. In particular, it may be worth investigating relations with Data Integration and Record Linkage algorithms.

Given the data lake and data exploration scenario, where a huge number of tables may be available, the problem of scalability should be considered.
Since the proposed approach performs a pair–wise processing of attributes in all tables, this may be an issue.

Although the paper has merits, it lacks strong originality and results.

After Rebuttal:
I acknowledge mine reading authors' response. I confirm my evaluation on the paper having some merits, but originality is difficult to assess since work on data integration (not necessarily by the semantic web community) is not taken into account sufficiently, as also reported by other authors. The main declared contribution, that is explaining the extracted relations, is actually not new in literature. Scalability is also a confirmed issue that is not considered in the current version of the paper.

**Anonymity:**

Yes, I would like my review to remain anonymous.

**Rating:**

-1: Weak Reject

**Reuse And Availability:**

2: Low

**Strong Points:**

- clear problem
- different approaches considered

**Subreviewer:**

I submitted this review.

**Weak Points:**

- related work missing related work in the field of databases
- lacking strong originality in the problem and in the proposed approaches
- results marginally better than previous solutions

---

> ### Author Rebuttal · Authors · 2021-01-29
>
> Here we provide some responses to the reviewer's comments:
>
> - "Related work missing related work in the field of databases."
> >*The related work section currently focuses on data discovery, to which we aim to contribute, and on language inference techniques, on which we build. Several reviewers would have liked us to consider other adjacent areas as well, in particular information integration, ontology alignment and semantic annotation. We are happy to extend the related work section to include these areas.
> We think that our work complements existing results in all these areas. In information integration, though we could provide an additional type of evidence to composite matchers, we note that in practice most data integration is highly manual, and thus our emphasis on explainability is relevant to users of data integration systems where users take fine-grained control over most decisions.  In relation to work on ontology alignment, we are trying to make minimal assumptions about the semantics available about the source and the target, while still inferring useful semantic relations between attributes. In relation to semantic annotation, in addition to the work of Jiaoyan Chen et al., there has been recent work evaluating the use of BERT as a supervised strategy (http://www.vldb.org/pvldb/vol13/p2549-li.pdf), to which we could easily relate our approach, though we note that our natural language inference angle seems to be distinctive.*
>
> - "Why is it focused on data lakes and not databases? it may be worth investigating relations with Data Integration and Record Linkage algorithms."
> >*Several reviewers have commented on our choice of data lakes as a motivating context. We are motivated by the rapid growth in data lakes, and associated growth in data catalogs to help users to come to terms with the available data (the Data Catalog market is worth several hundred million dollars and growing rapidly: https://www.marketsandmarkets.com/Market-Reports/data-catalog-market-48918216.html).  However, we agree with the reviewers that the techniques developed are not specific to data lakes, and are equally relevant to other sources of autonomously produced data. This is reflected in the evaluation, which makes use of open government data. We would be happy to emphasise this point in the paper, and even to change the title if the emphasis on data lakes is felt to be misleading.*
>
> - "Results marginally better than previous solutions"
> >*Indeed, empirically analyzing the similarity-based discovery capabilities of our proposal does result in marginally improved results over the ones from data discovery state-of-the-art. However, our central contribution focuses on the explainability dimension of these results, i.e., providing explicit characterisation of semantic relations between data sets. Data discovery state-of-the-art proposals do not present such a dimension, being limited to only quantifying a general notion of similarity between candidates (score-only). Thus, our approach matches the state-of-the-art effectiveness, while also providing explanations as to how an identified data set relates to a provided target.*
>
> - " Given the data lake and data exploration scenario, where a huge number of tables may be available, the problem of scalability should be considered."
> >*It is fair to say that the approach presented here would not scale directly to work with huge repositories, as it requires pairwise comparisons of the target with the sources.  However, it can be used in conjunction with index-based search strategies, such as D3L and Aurum, which would have the effect of pre-filtering candidate results.  This approach, in which semantic relations are inferred on the result of an indexing strategy is explored empirically in Section 4.3.*

---

### Decision · Program_Chairs · 2021-02-23

**Decision:**

Accept

**Comment:**

The work tackles the issue of finding semantically related attributes in a data lake and explaining those relations. It presents two approaches, one based on BERT, and another based on natural language inference over relational data. The study is anchored on a strong evaluation and results are positive.

**Strong Points:**
- interesting and relevant topic
- good evaluation design
- interesting methods

**Weak points:**
- description of approach lacks some details
- context w.r.t. related areas in SW could be improved
- a discussion of impact downstream tasks is missing